# Follow-Up of SARS-CoV-2 Antibody Levels in Belgian Nursing Home Residents and Staff Two, Four and Six Months after Primary Course BNT162b2 Vaccination

**DOI:** 10.3390/vaccines12080951

**Published:** 2024-08-22

**Authors:** Eline Meyers, Liselore De Rop, Fien Engels, Claudia Gioveni, Anja Coen, Tine De Burghgraeve, Marina Digregorio, Pauline Van Ngoc, Nele De Clercq, Laëtitia Buret, Samuel Coenen, Ellen Deschepper, Elizaveta Padalko, Steven Callens, Els Duysburgh, An De Sutter, Beatrice Scholtes, Jan Y. Verbakel, Stefan Heytens, Piet Cools

**Affiliations:** 1Department of Diagnostic Sciences, Faculty of Medicine and Health Sciences, Ghent University, 9000 Ghent, Belgium; eline.meyers@ugent.be (E.M.); fienengels@outlook.com (F.E.);; 2EPI-Centre, Department of Public Health and Primary Care, KU Leuven, 3000 Leuven, Belgium; 3Department of Public Health and Primary Care, Faculty of Medicine and Health Sciences, Ghent University, 9000 Ghent, Belgiumstefan.heytens@ugent.be (S.H.); 4Research Unit of Primary Care and Health, Department of General Medicine, Faculty of Medicine, University of Liège, 4000 Liège, Belgium; 5Department of Family Medicine & Population Health, Faculty of Medicine and Health Sciences, University of Antwerp, 2000 Antwerp, Belgium; 6Biostatistics Unit, Faculty of Medicine and Health Sciences, Ghent University, 9000 Ghent, Belgium; 7Laboratory of Medical Microbiology, Ghent University Hospital, 9000 Ghent, Belgium; 8Department of Internal Medicine & Infectious Diseases, Ghent University Hospital, 9000 Ghent, Belgium; 9Department of Epidemiology and Public Health, Sciensano, 1000 Brussels, Belgium

**Keywords:** SARS-CoV-2 antibodies, nursing home residents, nursing home staff, COVID-19 vaccination

## Abstract

When COVID-19 vaccines were implemented, nursing home residents (NHRs) and staff (NHS) in Belgium were prioritized for vaccination. To characterize the vaccine response over time in this population and to identify poorly responding groups, we assessed antibody concentrations two (T1), four (T2) and six months (T3) after primary course BNT162b2 vaccination in six groups of infection-naive/infection-primed NHRs/NHS, with/without comorbidity (NHRs only). Participant groups (N = 125 per group) were defined within a national serosurveillance study in nursing homes, based on questionnaire data. Dried blood spots were analyzed using ELISA for the quantification of SARS-CoV-2 S1RBD IgG antibodies. Among all groups, antibody concentrations significantly decreased between T1 and T2/T3, all with a ≥70% decrease at T3, except for infection-primed staff (−32%). Antibody concentrations among infection-naive NHRs were 11.96 times lower than those among infection-primed NHR, while the latter were comparable (x1.05) to infection-primed NHS. The largest proportion [13% (95% CI: 11–24%)] of vaccine non-responders was observed in the group of infection-naive NHRs with comorbidities. A longer interval between infection and vaccination (≥3 months) elicited higher antibody responses. Our data retrospectively show the necessity of timely COVID-19 booster vaccination. Infection-naive NHRs require special attention regarding immune monitoring in future epidemics or pandemics.

## 1. Background

During the first waves of the COVID-19 pandemic in Belgium, nursing home residents (NHRs) were at high risk of developing severe infection. Therefore, when vaccines were implemented in January 2021, NHRs were prioritized in the Belgian vaccination strategy. Vaccine rollout occurred quickly, as by the end of March 2021, 89.4% of NHRs and 76.8% of nursing home staff (NHS) in Belgium had been vaccinated with two doses of the BNT162b2 vaccine [1]. Although it was shown in clinical trials that the BNT162b2 COVID-19 vaccine was able to elicit robust immune responses in the general population, data on immune responses in an elderly population were limited, as they were underrepresented and generally healthy [2,3]. Additionally, only data for up to two months after vaccination were available at the moment of vaccine market authorization [3]. Nevertheless, older and frailer individuals, like NHRs, are known to suffer from immunosenescence (aging of the immune system) [4]. Moreover, since the marketing of the vaccine, it has been shown that COVID-19 vaccines lose efficacy over time and in the context of different variants [5]. This is especially true among the elderly, as it was shown that vaccine efficacy against infection dropped from 74.7% in March–May 2021 to 53.1% in June 2021 in this group [6]. Several countries adopted different vaccine strategies, with different intervals between vaccine doses and different groups prioritized [7]. Together with regional differences in viral transmission, this created unique epidemiological situations by country, increasing the importance of nationwide studies. A better understanding of the dynamics of immune responses in high-risk populations, like NHRs, is needed to establish uniform, efficient and evidence-based vaccination programs. Secondly, vulnerable groups should be identified to prioritize those who can benefit the most from repetitive immunization to allow for a tailored approach. Therefore, in the present study, we studied SARS-CoV-2 antibody concentrations in six pre-defined groups of NHRs and NHS in Belgium (SARS-CoV-2 infection-naïve/infection-primed and/or comorbidities) approximately two months (T1), four months (T2) and six months (T3) after primary course BNT162b2 vaccination.

## 2. Materials and Methods

### 2.1. Study Design

The data described in the current publication were collected within a national SARS-CoV-2 serosurveillance study among 1640 NHRs and 1368 NHS from 69 nursing homes in Belgium that was initiated at the moment of COVID-19 vaccine implementation (SCOPE study). The recruitment procedures within the SCOPE study have been described previously [8]. For the current study objectives, we defined six groups within the total study population, post hoc, based on the following participant characteristics: NHRs/NHS, SARS-CoV-2 infection-primed/infection-naive, comorbidity/no comorbidity (N = 125 per group). An overview of the participant groups and respective criteria is given in Table 1. Only participants who received primary course BNT162b2 vaccination were included. All participants from the SCOPE study who met the criteria for a respective group were listed, and 125 participants were randomly selected from the list per group. The sample size calculation is described in Section 3.1. Dried blood spots collected in April, June and August 2021 (T1, T2 and T3, respectively) from the selected subjects were analyzed to assess the concentration of SARS-CoV-2 antibodies. Figure 1 gives an overview of the timing of the primary vaccination campaign and sampling timepoints in the study.

### 2.2. Ethics

This study was approved by the Ethics Committee of the Ghent University Hospital (reference number BC-08719) and conducted according to the principles outlined in the Declaration of Helsinki. Each participant or their legal representative (e.g., for NHRs with dementia) signed an informed consent form after being informed about the goal of the study and the study procedures.

### 2.3. Data Collection

#### 2.3.1. Dried Blood Spot Collection

To assess SARS-CoV-2 IgG antibody concentrations, we collected dried blood spots (DBSs), which we previously validated as a valid specimen for (longitudinal) quantitative SARS-CoV-2 antibody detection [9,10]. Trained study staff obtained capillary blood with a fingerprick using an 18G lancet 1.8 mm depth (SARSTEDT Ag & Co., Nümbrecht, Germany) and collected it on a DBS saver card (EUROIMMUN, Lübeck, Germany). A minimum of four 6 mm diameter circles were filled with blood until saturation was visible on the back of the card and left to air dry for at least one hour. Samples were stored at −20 °C until analysis.

#### 2.3.2. Questionnaires

Sociodemographic and clinical data were collected using online questionnaires (LimeSurvey version 3.22, Hamburg, Germany). The nursing home head nurse(s) completed the questionnaires for NHRs. Participant characteristics [e.g., age, sex, care dependency level (for NHRs)] and COVID-19-relevant comorbidities (cardiovascular disease, diabetes mellitus, hypertension, immunodeficiency/-suppression, severe renal/lung/cardiac disease, active cancer) were recorded. Additionally, at every sample collection round, participants were asked about their COVID-19 vaccination status (number of doses, date of vaccination, vaccine type by brand name) and infection status (as assessed by self-reporting of previous PCR and/or antigen test and/or CT-scan, with the respective test results, and date of testing in case of a positive test result).

### 2.4. Quantification of SARS-CoV-2 S1RBD IgG Antibodies

All DBSs were analyzed using the SARS-CoV-2 anti-spike 1 receptor binding domain (S1RBD) IgG ELISA assay (ImmunoDiagnostics, Hong Kong) by trained laboratory staff. DBSs were processed as previously described [9]. First, a 6 mm diameter circle was punched with a single-hole puncher (Artemio, Braine-le-Comte, Belgium) from the DBS and placed in 250 µL 1× S1RBD IgG ELISA sample buffer (ImmunoDiagnostics, Hong Kong) in a U-shaped-bottom 96-wellplate. After one hour of incubation at 37 °C, the extract was diluted 100-fold, after which it was loaded onto the S1RBD-coated well plate (ImmunoDiagnostics, Hong Kong). The subsequent procedures were conducted manually, as described in the instruction manual of the ELISA assay [11]. Optical density (OD) was measured on the Behring ELISA Processor III (Siemens AG, Munich, Germany) at 450 nm. Extracts with an OD that exceeded the OD of the highest standard were re-analyzed using a 1000-fold dilution of the extract, and extracts with an OD lower than the OD of the lowest standard were re-analyzed using a non-diluted extract and 10-fold dilution of the extract. A set of SARS-CoV-2 antibody standards (ImmunoDiagnostics, Hong Kong) was used to generate a 4PL logistic regression curve to calculate the antibody concentrations (international units per ml, IU/mL) in GraphPad Prism 10 (GraphPad Software Inc., San Diego, CA, USA). The cutoff for SARS-CoV-2 seropositivity was 26 IU/mL [9].

### 2.5. Statistical Analysis 

Participants with a self-reported SARS-CoV-2 infection during the period of follow-up (between the second vaccine dose and sampling in August 2021) were excluded from the statistical analysis (S0: n = 2; S1; n = 1). This was carried out to exclude the effect of antibody boosting by natural infection during follow-up. We reported geometric mean antibody concentrations (GMC) with 95% confidence intervals per group, per timepoint. Normality was assumed for the log-transformed data (log10). We assessed differences in antibody concentrations between timepoints by means of a mixed effects model on the log-transformed data, with time as a fixed effect and individual and residual as random effects. Antibody concentrations measured at T1 were considered as the baseline, and differences over time were assessed by comparing to the respective previous timepoint and baseline. *p*-values were adjusted for multiplicity using Bonferroni’s correction. Additionally, we calculated the percentage decline with respective confidence intervals between timepoints. Differences in antibody concentration between groups were assessed by an unpaired *t*-test after log transformation of the data. *p*-values were adjusted for multiple testing using Bonferroni’s method. To compare antibody concentrations among groups, we divided the GMC of group B by group A per timepoint. An average GMC ratio was calculated by taking the mean of the GMC ratios for T1, T2 and T3. We reported the proportion of seronegative participants per group and per timepoint (with those at T1 considered as vaccine non-responders), with the 95% CI calculated by the Wilson–Brown method.

Exploratively, in infection-primed NHRs and NHS (R1 and S1), we assessed whether the time between previous SARS-CoV-2 infection and vaccination impacted the SARS-CoV-2 antibody concentrations after vaccination. Group R1c was not considered, to avoid potential bias from comorbidity as a predictor for SARS-CoV-2 antibody concentration. Two groups per participant type were defined: NHRs/NHS with an infection <90 days but >14 days before COVID-19 vaccination, and NHRs/NHS with an infection ≥90 days before COVID-19 vaccination. The cutoff of 90 days was based on previous research [12]. Differences in SARS-CoV-2 antibody concentration between the two groups were assessed by means of an unpaired *t*-test after log transformation of the data, and *p*-values were adjusted for multiplicity using Bonferroni’s method. 

Additionally, exploratively, we assessed the associations between comorbidity (type and number) and S1RBD IgG response at T1–T3 by fitting mixed effects models using the data from groups R0c and R1c. A univariate analysis was performed, with S1RBD IgG as the dependent variable and comorbidity (type/number), previous infection status, time since vaccination (T1/T2/T3) and the interaction terms of previous infection status and time since vaccination and type of comorbidity as fixed effects. Time since vaccination was also included as a repeated effect. Variable selection was based on the results from longitudinal follow-up between groups. Adjusted two-sided *p*-values of ≤0.05 were considered statistically significant. All statistical analyses were performed using Graphpad Prism 9 (GraphPad Software Inc., San Diego, CA, USA), except for the mixed effect model analysis for type/number of comorbidities, which was performed using IBM SPSS Statistics Version 28.0 (IBM Corp., New York, NY, USA).

## 3. Results

### 3.1. Sample Size

As we initially planned to report SARS-CoV-2 antibody concentrations at an optical density (OD) ratio, we calculated the sample size estimation to detect a mean difference between the groups at an OD ratio of 2, assuming a standard deviation of 3.8 (Cohen’s d = 0.53), power of 80% and a significance level of 1% (two-sample *t*-test), resulting in a minimum of 86 subjects per group. To compare concentrations between follow-up visits, we calculated the sample size to detect a mean difference of 0.5 (Cohen’s d), assuming a standard deviation of 1, a power of 80% and a significance level of 5%, resulting in a minimum of 51 subjects per group. As a drop-out rate of 30% was considered over a 10-month period, we augmented the minimum sample size per group up to n = 123 [86/(1 – 0.3)]. Due to the availability of a commercial kit reporting in international units/mL, we opted for a quantitative unit instead of the OD ratio.

### 3.2. Participant Characteristics

Table 2 gives an overview of the median age, sex and type of comorbidity (if applicable) per participant group, as well as the number of available samples per timepoint. In the different NHR and NHS groups, the median age was similar, ranging between 86 and 89 years and between 43 and 44 years, respectively. The percentage of females was significantly higher than that of males across groups, and highest in the NHS groups. In the two NHR groups with comorbidities, cardiovascular disease was the most prevalent comorbidity, followed by hypertension. The prevalences of immunodeficiency/immunosuppression and cancer were below 5% in the two groups.

### 3.3. S1RBD IgG Antibody Concentrations in NHRs and NHS at T1, T2 and T3

#### 3.3.1. Significantly Waning S1RBD IgG Antibody Concentrations Post-Vaccination

The (geometric mean) S1RBD IgG antibody concentrations over time per participant group are visualized in Figure 2. At both T2 and T3, mean antibody concentrations significantly decreased in all participant groups compared to T1. Only in infection-primed NHS did the mean antibody concentration initially significantly drop at T2 but remain stable between T2 and T3. Accordingly, the smallest decline (−32%, 95% CI −32%; −33%) in antibody concentration at T3 was observed among infection-primed NHS, as for all other groups the antibody concentrations decreased by 53–79% at T2 and 74–87% at T3, compared to baseline (see Figure 2). The highest drop in antibody concentration was observed among infection-naive NHS, with a decrease of 87% (95% CI −87%; −86%) at T3. Moreover, among NHRs, several did not show an antibody response to vaccination at T1 (vaccine non-responders) or had rapidly waning antibody responses decreasing below the level for seropositivity at T2 or T3. The largest proportion of seronegative participants was observed in the group of infection-naive NHRs with ≥1 comorbidity, as 13% (95% CI: 11–24%) did not have a detectable antibody response (non-responders) at T1, leading up to 36% (95% CI: 28–47%) at T3. In comparison, none of the NHS tested seronegative across timepoints. Table 3 gives an overview of the proportion of seronegative participants per group and per timepoint.

#### 3.3.2. Differences in S1RBD IgG Antibody Concentration Post-Vaccination between Infection-Naive and Infection-Primed NHRs and NHS, with and without Comorbidity

Table 4 gives an overview of the geometric mean antibody concentrations per group and adjusted *p*-values to assess statistically significant differences in antibody concentration between groups. For all timepoints, across all groups, the lowest S1RBD IgG mean antibody concentrations were observed in infection-naive NHRs with ≥1 comorbidity, followed by infection-naive NHRs without comorbidity.

##### NHRs vs. NHS

Antibody concentrations were compared between NHRs and NHS, stratified for infection status (infection-primed/infection-naive). We found that infection-naive NHRs had 3.19 times lower geometric mean antibody concentrations compared to infection-naive NHS, averaged over all timepoints (*p* < 0.001 at all timepoints). Contrastingly, infection-primed NHRs initially had higher antibody concentrations (T1) than infection-primed NHS (*p* < 0.05); however, the difference in mean GMC diminished to comparable mean antibody concentrations at T2 and T3 (*p* > 0.05). The overall ratio of geometric mean antibody concentrations between infection-primed NHRs and NHS was 1.05. Compared to infection-naive NHS, NHRs who were infection-primed had significantly higher (x 3.94) antibody concentrations (*p* < 0.001 at all timepoints).

##### SARS-CoV-2 Infection-Naive vs. Infection-Primed NHRs/NHS

Infection-naive NHRs showed a poor SARS-CoV-2 antibody response to vaccination, with 11.57 times lower geometric mean antibody concentrations compared to infection-primed NHRs (*p* < 0.001 at all timepoints). Infection-naive NHS had generally 4.55 times lower antibody levels compared to infection-primed NHS. The difference between these groups was most prominent at T2 and T3 (*p* = 0.054 at T1, *p* < 0.001 at T2 and T3). 

##### Comorbidity vs. No Comorbidity

No statistically significant differences in antibody concentration were observed between NHRs with and without comorbidity, stratified for infection status (*p* > 0.05 at all timepoints). 

### 3.4. SARS-CoV-2 Antibody Concentrations after Vaccination Stratified According to Time since Last SARS-CoV-2 Infection

As NHRs and NHS who were naturally infected prior to vaccination had significantly higher S1RBD IgG antibody concentrations compared to infection-naive NHRs/NHS, we explored whether the interval between natural infection and vaccination (administration of the first dose) affected post-vaccine antibody levels. Table 5 gives an overview of the geometric mean S1RBD IgG antibody concentrations among NHRs/NHS with a prior natural infection <3 months but >14 days before vaccination versus 3–10 months prior to vaccination. All infection-primed NHRs/NHS reported only a single SARS-CoV-2 infection before vaccination.

A consistent trend of higher antibody concentrations in case of older infections was seen across different months post-vaccination and populations. NHS who were naturally infected shortly before vaccination (<3 months, >14 days) had statistically significantly lower antibody responses at T1 and T2 compared to NHS who had an infection 3–10 months prior to vaccination. Similarly, at T3, antibody concentrations in NHRs were statistically significantly lower among NHRs who were naturally infected shortly before vaccination, compared to NHRs who were infected in the 3–10 months prior to vaccination. 

### 3.5. Effect of Association between Type and Number of Comorbidities on SARS-CoV-2 Antibody Concentrations after Vaccination

As no statistically significant differences were found between NHRs with ≥1 and without comorbidity (see Table 4), we performed mixed effects models to assess associations between the type and number of comorbidities and SARS-CoV-2 antibody response post-primary vaccination. The mean estimates (95% CI) and significance values for the association between the type/number of comorbidities and (the decline in) S1RBD IgG antibody concentrations are presented in Appendix A. Our observations suggest that NHRs with hypertension had a lower S1RBD IgG antibody response after primary course vaccination compared to NHRs without hypertension (*p* = 0.014). Additionally, we observed a faster decline in antibody concentrations in NHRs without cardiovascular disease (*p* = 0.015) and generally lower antibody concentrations in NHRs without severe cardiac/pulmonary/renal disease (*p* = 0.028), compared to NHRs with the condition. The number of comorbidities was not associated with SARS-CoV-2 antibody response.

## 4. Discussion

In the current study, we assessed SARS-CoV-2 S1RBD IgG antibody responses two (T1), four (T2) and six months (T3) after primary course BNT162b2 vaccination in infection-naive/infection-primed NHRs/NHS with or without comorbidities (the latter for NHRs only). We found that antibody concentrations in all groups significantly decreased within four months after vaccination, compared to baseline. Moreover, antibody concentrations further significantly decreased between month four and month six in all groups, except for infection-primed NHS. The decline in antibody concentration at month six post-vaccination was similar across groups (−74; −86%), except for infection-primed NHS, where a smaller decrease was observed (−32%; 95% CI −32%; −33%). The largest proportion of participants testing seronegative post-vaccination was observed among infection-naive NHRs (with/without ≥1 comorbidity). Similarly, the lowest antibody concentrations across all timepoints were observed in these groups. No difference in antibody response was observed in NHRs with vs. without comorbidity. Additionally, we found that having a natural infection ≥3 months before vaccination resulted in significantly higher post-vaccine antibody responses compared to naturally acquired infections shortly (<3 months) before vaccination.

Although we observed a significant waning of antibodies among all participant groups within four and six months post-vaccination compared to baseline, our data clearly identified a poorly responding group, which was infection-naive NHRs. In contrast to infection-primed NHRs, antibody concentrations at T1 were 11.96 lower among infection-naive NHRs, meaning that not all NHRs had a poor response to vaccination. In contrast, infection-primed NHRs developed antibody concentrations comparable to those of younger-aged equals (infection-primed NHS) or even higher compared to infection-naive NHS. Others have previously similarly identified infection-naive NHRs as poor responders after primary course COVID-19 vaccination [13,14,15,16,17]. Our longitudinal data show that, six months post-vaccination, mean geometric antibody concentrations decreased to as low as 49.74 IU/mL (95% CI 34.87–70.96) and 81.71 IU/mL (95% CI 61.35–108.80) among infection-naive NHRs with ≥1 comorbidity and without comorbidity, respectively. As antibody concentrations among these groups waned to critical levels, 36% (95% CI 28–47) and 17% (95% CI 11–25) of infection-naive NHRs tested seronegative six months after vaccination among infection-naive NHRs with ≥1 comorbidities and without comorbidities, respectively. Even four months after vaccination, antibody levels decreased by >50% among all groups except for infection-primed staff. As anti-spike SARS-CoV-2 antibodies are repeatedly suggested as a correlate for COVID-19 protection, it is likely that clinical protection is also affected in these groups [18,19,20]. As an example, a study from Triebelhorn et al. showed that individuals lacking a humoral response after vaccination (non-responders) had much higher re-infection rates (14.6%) compared to responders (2.1%). However, no consensus on an immunological cut-off level for COVID-19 protection has been reached, which impedes straightforward advice regarding the timing of (booster) vaccination. Some studies have previously suggested empirical antibody threshold levels; however, reporting in different units hinders comparability [19,21,22]. 

In Belgium, approximately ten months after primary course vaccination, a COVID-19 booster campaign was initiated among NHRs. Rapidly thereafter, NHS were also administered booster doses. Retrospectively examined, our data show that booster vaccination among infection-naive NHRs was critical at the time, as the humoral immune response was no longer safeguarded. This was especially true since, at that time, the SARS-CoV-2 Omicron variant was rapidly emerging, and was shown to be more transmissible than previous circulating strains [23]. Others have indeed shown that repeated immunization by means of COVID-19 vaccine booster doses elicits more robust and persistent antibody responses in NHRs, decreasing the number of seronegatives [24,25]. However, apart from humoral immune responses, other mechanisms exist through which vaccination can provide protection against COVID-19, like cellular immunity [26]. Specifically, T cell immunity is suggested to be particularly important for protection against COVID-19 in individuals with hampered humoral immunity [27,28,29]. Moreover, it is suggested that T cell immunity plays an important role in protection against severe COVID-19 disease, even in the absence of humoral response [30]. Indeed, despite significantly waning antibody responses after primary course vaccination, only three subjects (NHS) in the participant subset reported a breakthrough infection during follow-up. Generally, in Belgium, the weekly COVID-19 incidence among NHRs remained below 0.15 lab-confirmed cases per 100 NHRs in the six months after the completion of the primary course vaccination campaign. From this perspective, it is debatable that clinical protection against COVID-19 is safeguarded by other mechanisms, despite waning SARS-CoV-2 antibody levels [31]. 

Nevertheless, based on our findings, special attention should be given to infection-naive NHRs with regard to vaccination against other infectious diseases or in future epidemics or pandemics. The reduced and rapidly waning immune responses highlight the necessity for proactive and sustained immunization strategies to ensure optimal protection, which can be tailored to specific high-risk subgroups, like infection-naïve NHRs. 

Our findings regarding the interval between previous infection and vaccination are supported by Anichini et al. [12]. These authors also found that, compared to a 1-to-2-month interval, a 2-to-3-month interval between a preceding SARS-CoV-2 infection and vaccination elicited higher antibody responses. Tuaillon et al. and Stellini et al. showed that post-vaccine antibody levels were higher for those previously infected in the first epidemic wave than those in the second epidemic wave, corresponding to 9-to-12-month and 3-to-7-month intervals between infection and vaccination, respectively [32,33]. Analogously, it has also been reported that an extended interval between the first and second doses of COVID-19 vaccination elicited higher SARS-CoV-2 antibody responses compared to the standard BNT162b2 regimen [34,35]. This might be explained by the enhanced differentiation of B cells into plasmablasts with an extended dosing interval, mediated by interleukin 2 (IL-2), as suggested by Payne et al. [35]. Alternatively, the observed differences in antibody response associated with the interval between infection and vaccination could also be due to differential immunogenicity to different viral strains or different assay sensitivity to antibodies elicited by different viral strains. Among our study population, >3 months in advance of primary course vaccination (first wave), the wild-type (Wuhan) strain was the predominant strain, while in the <1 month prior to the vaccination campaign (second wave), the alpha (British) strain started to become prevalent [31]. In relation to this, it is important to note that our assay detected SARS-CoV-2 antibodies binding to wild-type anti-spike antigen, and hypothetically, detection may not therefore be equally sensitive across antibodies directed against different viral strains. This could potentially create a bias in comparing antibody levels that are elicited by different SARS-CoV-2 viral strains. As our findings are solely observational, more research is needed to explore the optimal interval between natural infection and vaccination to enhance antibody responses.

Exploratively, we investigated the associations between the type and number of comorbidities and the S1RBD IgG response post-vaccination. Our data suggest that hypertension was associated with lower vaccine antibody levels in NHRs. These findings are also suggested by previous papers on the association between high blood pressure and SARS-CoV-2 antibody levels after vaccination [36,37,38]. Prior studies have already demonstrated the link between hypertension and immune system regulators, like deficiency in T and B cells and increased levels of proinflammatory cytokines [39,40]. The latter are known to be associated with immunosenescence and inflammaging (age-related increases in chronic inflammatory status) [41,42,43]. Additionally, antihypertensive medication, like ACE-inhibitors, were shown to have immunomodulating properties, which also might contribute to the observed association between hypertension and post-COVID-19-vaccination S1RBD IgG responses [44,45]. Counterintuitively, our analysis showed that cardiovascular disease was associated with a smaller decrease in humoral vaccine response, and severe cardiac/pulmonary/renal disease was associated with higher post-vaccination antibody concentrations. However, here, the literature reports conflicting findings, as chronic cardiovascular, pulmonary and renal diseases tend to be associated with lower vaccine immunogenicity [46,47,48,49]. However, the current study was not designed to assess associations with different comorbidities; hence, the results can only be suggestive and should be interpreted with caution.

### 4.1. Limitations

Although important observations have been made regarding the follow-up of humoral immune responses six months post-vaccination in a large sample of NHRs and NHS, some limitations should be addressed. Firstly, the time between vaccination and sampling differed between participants, as sampling occurred independently from the vaccination date. Nevertheless, the median number of days (with interquartile range) between the second vaccine dose and T1, T2 and T3 approximated two-month intervals: 67 days (64–76); 127 days (123–135); 188 days (183;195), respectively.

Secondly, the categorization of infection-naive/infection-primed participants relied on the self-reporting of a positive RT-PCR/antigen/CT-scan test result in a two-monthly survey. Therefore, the proportion of infection-primed participants could be underestimated due to misreporting, asymptomatic cases, shortage/lacking test capacity and/or missing questionnaire responses.

### 4.2. Conclusions

In the current study, we investigated SARS-CoV-2 antibody responses among six pre-defined groups of infection-naive/infection-primed NHRs/NHS with/without comorbidity two, four and six months after BNT162b2 primary course vaccination. We found that antibody concentrations significantly decreased within four months after vaccination among all groups, and further decreased at six months except among infection-primed NHS. Infection-naive NHRs in particular had poor antibody concentrations after primary course vaccination, with a large proportion testing seronegative after vaccination, especially those suffering from ≥1 comorbidity. Contrastingly, infection-primed NHRs had antibody levels similar to younger-aged equals (infection-primed NHS). Moreover, we found that a longer interval (≥3 months) between natural infection and vaccination yielded higher antibody concentrations compared to natural infection shortly before vaccination. These data retrospectively show the necessity of timely COVID-19 booster vaccination, since antibody concentrations after primary course BNT162b2 vaccination rapidly waned within six months. In future pandemic or epidemic settings, infection-naive NHRs should be given special attention with regard to immune monitoring and vaccine strategies.

## Figures and Tables

**Figure 1 vaccines-12-00951-f001:**
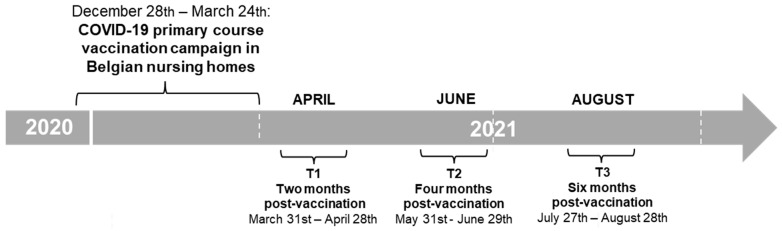
Overview of the timing of the primary course vaccination campaign in Belgian nursing homes and sampling timepoints in the study (approximately two months (T1), four months (T2) and six months (T3) after vaccination). Vertical white dashed lines indicate three-month timespans.

**Figure 2 vaccines-12-00951-f002:**
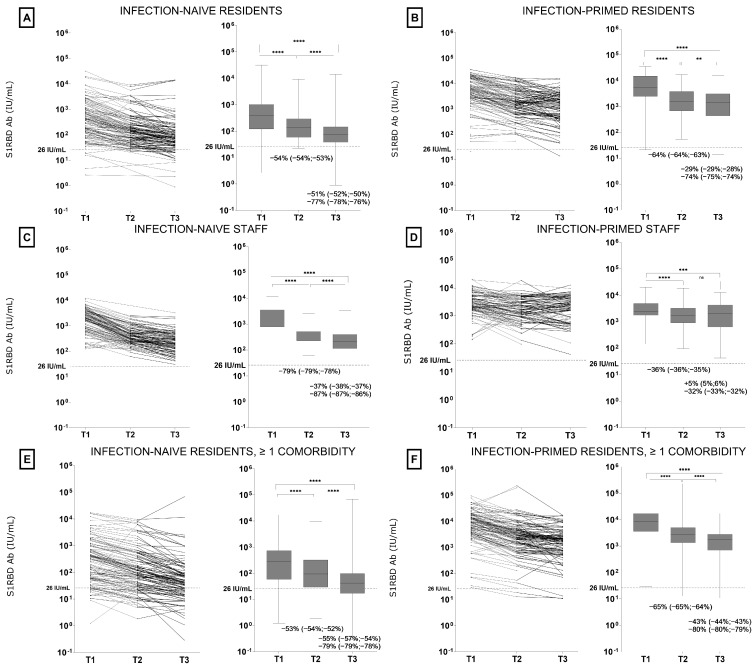
S1RBD IgG antibody concentrations in nursing home residents and staff two (T1), four (T2) and six months (T3) after two-dose regimen BNT162b2 vaccination. S1RBD IgG antibody concentrations over time are visualized per sub-cohort by spaghetti plots (**left**) and box plots (**right**). Panel (**A**): Infection-naive residents, no comorbidity. Panel (**B**): Infection-primed residents, no comorbidity. Panel (**C**): Infection-naive staff, no comorbidity. Panel (**D**): Infection-primed staff, no comorbidity. Panel (**E**): Infection-naive residents, ≥1 comorbidity. Panel (**F**): Infection-primed residents, ≥1 comorbidity. **** = adjusted *p* ≤ 0.0001, *** = adjusted *p* ≤ 0.001, ** = adjusted *p* ≤ 0.01, ns = adjusted *p* > 0.05. Percentages with 95% confidence intervals below the boxplot indicate the decrease in geometric mean antibody concentration between T1 and T2 post-vaccination (below middle boxplot), between T2 and T3 (upper value below right boxplot) and between T1 and T3 (lower value below right boxplot). Lower and upper error bars represent the minimum and maximum S1RBD IgG concentrations, respectively.

**Table 1 vaccines-12-00951-t001:** Overview of the participant groups defined in the current study. N, number.

Abbreviation	Participant Group Criteria	N
R	Residents	
R0	No comorbidity, infection-naive	125
R1	No comorbidity, infection-primed	125
R0c	Comorbidity (≥1), infection-naive	125
R1c	Comorbidity (≥1), infection-primed	125
S	Staff	
S0	No comorbidity, infection-naive	125
S1	No comorbidity, infection-primed	125

**Table 2 vaccines-12-00951-t002:** Median age, sex, type of comorbidity per participant group and number of available samples per group and timepoint.

Participant Group	Median Age, Years (IQR)	Female (%)	CVD (%)	DM (%)	HTN (%)	RLC (%)	ID (%)	Cancer (%)	# Available Samples		
									T1	T2	T3
R0	89 (83–92)	82	NA	NA	NA	NA	NA	NA	124	100	112
R1	86 (80–91)	74	NA	NA	NA	NA	NA	NA	124	99	100
R0c	89 (85–92)	70	57	22	49	13	4	4	124	124	103
R1c	88 (83–92)	76	69	24	41	18	4	2	123	115	102
S0	44 (33–55)	86	NA	NA	NA	NA	NA	NA	125	96	88
S1	43 (35–50)	92	NA	NA	NA	NA	NA	NA	118	90	78

Participant groups: R0: infection-naive nursing home residents, no comorbidity; R1: infection-primed nursing home residents, no comorbidity; R0c: infection-naive nursing home residents, ≥1 comorbidity, R1c: infection-primed nursing home residents, ≥1 comorbidity; S0: infection-naive nursing home staff, no comorbidity; S1: infection-primed nursing home staff, no comorbidity. Comorbidities: CVD: cardiovascular disease; DM: diabetes mellitus; HTN: hypertension; RLC: severe renal, lung or cardiac disease; ID: immunodeficiency or immunosuppression. Timepoints: T1, two months after COVID-19 vaccination; T2, four months after COVID-19 vaccination; T3, six months after COVID-19 vaccination. IQR, interquartile range; NA, not applicable.

**Table 3 vaccines-12-00951-t003:** Proportion of seronegative subjects two (T1), four (T2) and six months (T3) post two-dose regimen BNT162b2 vaccination.

Participant Group	Seronegative T1 % (95% CI) (Vaccine Non-Responders)	Seronegative T2 % (95% CI)	Seronegative T3 % (95% CI)
R0	4 (2–9)	3 (1–8)	17 (11–25)
R1	1 (0–4)	0 (0–3)	1 (0–5)
R0c	13 (11–24)	22 (16–31)	36 (28–47)
R1c	0 (0–3)	2 (0–6)	3 (1–8)
S0	0 (0–3)	0 (0–4)	0 (0–4)
S1	0 (0–3)	0 (0–4)	0 (0–5)

Participant group: R0: infection-naive residents, no comorbidity; R1: infection-primed residents, no comorbidity; R0c: infection-naive residents, ≥1 comorbidity; R1c: infection-primed residents, ≥1 comorbidity; S0: infection-naive staff, no comorbidity; S1: infection-primed staff, no comorbidity. CI: confidence interval.

**Table 4 vaccines-12-00951-t004:** Geometric mean S1RBD IgG antibody concentrations across different participant groups two (T1), four (T2) and six months (T3) after BNT162b2 COVID-19 vaccination. GMC: geometric mean S1RBD IgG antibody concentrations. IU/mL: international units/mL.

	GMC^x^ (IU/mL)	95% Confidence Interval (IU/mL)	GMC^xx^ (IU/mL)	95% Confidence Interval (IU/mL)	Adjusted *p*-Value	Ratio GMC ^a^	Averaged Ratio GMC ^b^
Nursing home residents vs. nursing home staff							
	A. Infection-naive residents (R0)		B. Infection-naive staff (S0)			B/A	
T1	358	266–481	1685	1404–2022	<0.001	4.71	3.19
T2	166	127–217	354	308–407	<0.001	2.13	
T3	82	61–109	223	183–271	<0.001	2.73	
	C. Infection-primed residents (R1)		D. Infection-primed staff (S1)			D/C	
T1	4279	3284–5577	2511	2087–3021	0.029	0.59	1.05
T2	1543	1194–1993	1612	1315–1975	0.999	1.04	
T3	1100	836–1446	1698	1321–2182	0.516	1.54	
	B. Infection-naive staff (S0)		C. Infection-primed residents (R1)				
T1	1685	1404–2022	4279	3284–5577	<0.001	2.54	3.94
T2	354	308–407	1543	1194–1993	<0.001	4.35	
T3	223	183–271	1100	837–1446	<0.001	4.93	
SARS-CoV-2 infection-naive vs. infection-primed NHRs/NHS							
	A. Infection-naive residents (R0)		C. Infection-primed residents (R1)			C/A	
T1	358	266–481	4279	3284–5577	<0.001	11.96	11.57
T2	166	127–217	1543	1194–1993	<0.001	9.30	
T3	82	61–109	1100	837–1446	<0.001	13.46	
	B. Infection-naive staff (S0)		D. Infection-primed staff (S1)			D/B	
T1	1685	1404–2022	2511	2087–3021	0.054	1.49	4.55
T2	354	308–407	1612	1315–1975	<0.001	4.55	
T3	223	183–271	1698	1321–2182	<0.001	7.62	
No comorbidity vs. ≥1 comorbidity							
	A. Infection-naive residents (R0)		E. Infection-naive residents >1 comorbidity (R0c)			E/A	
T1	358	266–481	236	169–331	0.999	0.66	0.65
T2	166	127–217	111	80–153	0.999	0.69	
T3	81	61–109	50	35–71	0.651	0.61	
	C. Infection-primed residents (R1)		F. Infection-primed residents >1 comorbidity (R1c)			F/C	
T1	4279	3284–5577	6590	5070–8571	0.476	1.54	1.01
T2	1543	1194–1993	2329	1746–3107	0.796	1.51	
T3	1100	837–1446	1322	996–1755	0.999	1.20	

^a^ Ratio of geometric mean concentrations per timepoint (GMC^xx^/GMC^x^); ^b^ average ratio of geometric mean concentrations of all timepoints. Underligned *p*-values are statistically significant.

**Table 5 vaccines-12-00951-t005:** S1RBD IgG antibody concentrations two (T1), four (T2) and six (T3) months post-vaccination among nursing home residents and nursing home staff with natural infection <3 months prior to vaccination versus 3–10 months prior to vaccination.

	Geometric Mean Concentration (IU/mL)	95% Confidence Interval (IU/mL)	Geometric Mean Concentration (IU/mL)	95% Confidence Interval (IU/mL)	Adjusted *p*-Value (IU/mL)
	NHRs ^a^ with infection <3 months prior to vaccination ^b,c^ (n = 92)		NHRs ^a^ with infection 3–10 months prior to vaccination ^b,c^ (n = 31)		
T1	3817	2782–5236	7033	4625–10,697	0.274
T2	1369	1017–1843	2959	2009–4357	0.101
T3	872	629–1209	2337	1592–3430	0.005
	NHS with infection <3 months prior to vaccination ^b,c^ (n = 74)		NHS with older infection 3–10 months to prior vaccination ^b,c^ (n = 48)		
T1	1984	1549–2540	3559	2743–4617	0.013
T2	1213	946–1556	2300	1682–3145	0.010
T3	1265	909–1759	2380	1620–2380	0.078

^a^ Only the group of infection-primed residents without comorbidities was considered (R1). ^b^ Participants with a natural infection ≤14 days prior to vaccination were excluded from the analysis. ^c^ Interval was calculated from the number of days between confirmation of the SARS-CoV-2 infection and administration of the first vaccine dose. NHRs: nursing home residents; NHS: nursing home staff. IU/mL: international units/mL. Underlined *p*-values are statistically significant.

## Data Availability

The sponsor of this study (Sciensano) shares ownership of the data. Therefore, the data are available upon request and after permission of the sponsor.

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
