# Peer review of "Follow-Up of SARS-CoV-2 Antibody Levels in Belgian Nursing Home Residents and Staff Two, Four and Six Months after Primary Course BNT162b2 Vaccination"

_vaccines, 2024, doi:10.3390/vaccines12080951_

Round 1
Reviewer 1 Report
Comments and Suggestions for Authors
This is a useful compilation and analysis of data. However, the paper would have gained in value by a more detailed discussion on the possibility to use the findings to support vaccination policies. This would require a more in-depth discussion on the correlates of protection. There is a correlation between neutralizing and RBD-binding antibodies and protection against infection, but there are no commonly accepted threshold values. In addition, protection against severe disease persists in the absence of antibodies. T cells seem to be responsible for this sustained protection. Memory B cells may also play a role. I recommend to address these aspects in a new version of the discussion.
Reviewer 2 Report
Comments and Suggestions for Authors
1.It is inevitable for antibody titers to decrease over time after vaccination. The title of the paper does not reflect the research purpose.
2.The research investigated two populations: NHR and NHS. But the keywords only display NHR.
3.Line 37: (13% (95% CI:11-24%)) .Suggest using different parentheses.
4.Suggest placing the calculation of sample size in 2.1 Study Design.
5.Table3(LIne236-240):Please do not put the annotation content of the table together with the table title.
6.The references are not standardized. How many authors should be written?
7.All tables should be standardized.
Comments on the Quality of English Language
Minor editing of English language required.
Reviewer 3 Report
Comments and Suggestions for Authors
This manuscript reports on the temporal changes of IgG antibodies in the serum of Belgian NHR and NHS populations after BNT162b2 vaccination, which has certain reference value for exploring the protective effect of COVID-19 vaccine.
The detection method and equipment of this manuscript need further detailed description.
The innovation of the content in this manuscript needs to be further emphasized.
Reviewer 4 Report
Comments and Suggestions for Authors
This study evaluates the humoral response to the mRNA SARS-CoV-2 vaccine and the effect of a previous natural infection. Interestingly, the authors have been able to analyze and compare two different populations from the same environment, one composed of rather young and healthy individuals, the other composed of more elderly and fragile individuals. The statistical analyses sound appropriate and have been corrected for multiple testing. The paper is clearly written and the data are rather convincing. This work improves our knowledge of the factors influencing the response to SARS-CoV-2 vaccines and confirm the importance of booster vaccinations especially in the most fragile populations.
Here are a few minor points:
Line 53: “only data up to two months after vaccination”
Line 279 and 291: It should be written Table 5 not 4
Line 345: “antibody levels decreased by >50%”
Line 354: “…was shown to be more transmissible”
Round 2
Reviewer 3 Report
Comments and Suggestions for Authors
The results of sample size calculation can be shown in result section.
The incidence of COVID19 in NHR and NHS after vaccination or during the vaccinations should be presented in the manuscript. And the protecting effeciency of the vaccination and its relationship with the level of antibodies should be discussed in the manuscript.
Author Response
Comment 1: The results of sample size calculation can be shown in result section.
Response 1: We have moved it to the results section (line 181-193).
Comment 2: The incidence of COVID19 in NHR and NHS after vaccination or during the vaccinations should be presented in the manuscript. And the protecting effeciency of the vaccination and its relationship with the level of antibodies should be discussed in the manuscript.
Response 2: We agree with the interviewer that this information adds more depth to the discussion. We elaborated on this in line 358-387.